## EMERGING TECHNOLOGY

# Concerning RNA-guided gene drives for the alteration of wild populations

**Abstract** Gene drives may be capable of addressing ecological problems by altering entire populations of wild organisms, but their use has remained largely theoretical due to technical constraints. Here we consider the potential for RNA-guided gene drives based on the CRISPR nuclease Cas9 to serve as a general method for spreading altered traits through wild populations over many generations. We detail likely capabilities, discuss limitations, and provide novel precautionary strategies to control the spread of gene drives and reverse genomic changes. The ability to edit populations of sexual species would offer substantial benefits to humanity and the environment. For example, RNA-guided gene drives could potentially prevent the spread of disease, support agriculture by reversing pesticide and herbicide resistance in insects and weeds, and control damaging invasive species. However, the possibility of unwanted ecological effects and near-certainty of spread across political borders demand careful assessment of each potential application. We call for thoughtful, inclusive, and well-informed public discussions to explore the responsible use of this currently theoretical technology.

**KEVIN M ESVELT\*, ANDREA L SMIDLER, FLAMINIA CATTERUCCIA\* AND GEORGE M CHURCH\***

**\*For correspondence:**
kevin.esvelt@wyss.harvard.edu (KME); fcatter@hsph.harvard.edu (FC); gmc@harvard.edu (GMC)

**Reviewing editor**: Diethard Tautz, Max Planck Institute for Evolutionary Biology, Germany

## Introduction

Despite numerous advances, the field of molecular biology has often struggled to address key biological problems affecting public health and the environment. Until recently, editing the genomes of even model organisms has been difficult. Moreover, altered traits typically reduce evolutionary fitness and are consequently eliminated by natural selection. This restriction has profoundly limited our ability to alter ecosystems through molecular biology.

If we could develop a general method of ensuring that engineered traits would instead be favored by natural selection, then those traits could spread to most members of wild populations over many generations. This capability would allow us to address several major world problems, including the spread of insect-borne diseases, the rise of pesticide and herbicide resistance, and the

agricultural and environmental damage wrought by invasive species.

Scientists have long known of naturally occurring selfish genetic elements that can increase the odds that they will be inherited. This advantage allows them to spread through populations even if they reduce the fitness of individual organisms. Many researchers have suggested that these elements might serve as the basis for 'gene drives' capable of spreading engineered traits through wild populations (*Craig et al., 1960*; *Wood et al., 1977*; *Sinkins and Gould, 2006*; *Burt and Trivers, 2009*; *Alphey, 2014*). Austin Burt was the first to propose gene drives based on site-specific 'homing' endonuclease genes over a decade ago (*Burt, 2003*). These genes bias inheritance by cutting the homologous chromosome, inducing the cell to copy them when it repairs the break. Several efforts have focused

on the possibility of using gene drives targeting mosquitoes to block malaria transmission (*Scott et al., 2002*; *Windbichler et al., 2007, 2008, 2011*; *Li et al., 2013a*; *Galizi et al., 2014*). However, development has been hindered by the difficulty of engineering homing endonucleases to cut new target sequences (*Chan et al., 2013a*; *Thyme et al., 2013*; *Takeuchi et al., 2014*). Attempts to build gene drives with more easily retargeted zinc-finger nucleases and TALENs suffered from instability due to the repetitive nature of the genes encoding them (*Simoni et al., 2014*).

The recent discovery and development of the RNA-guided Cas9 nuclease has dramatically enhanced our ability to engineer the genomes of diverse species. Originally isolated from 'CRISPR' acquired immune systems in bacteria, Cas9 is a non-repetitive enzyme that can be directed to cut almost any DNA sequence by simply expressing a 'guide RNA' containing that same sequence. In little more than a year following the first demonstrations in human cells, it has enabled gene insertion, deletion, and replacement in many different species (*Bassett et al., 2013*; *Cho et al., 2013*; *Cong et al., 2013*; *DiCarlo et al., 2013*; *Friedland et al., 2013*; *Gratz et al., 2013*; *Hu et al., 2013*; *Hwang et al., 2013*; *Jiang et al., 2013a, 2013b*; *Jinek et al., 2013*; *Li et al., 2013b*; *Mali et al., 2013c*; *Tan et al., 2013*; *Upadhyay et al., 2013*; *Wang et al., 2013b*).

Building RNA-guided gene drives based on the Cas9 nuclease is a logical way to overcome the targeting and stability problems hindering gene drive development. Less obvious is the extent to which the unique properties of Cas9 are well-suited to overcoming other molecular and evolutionary challenges inherent to the construction of safe and functional gene drives.

We submit that Cas9 is highly likely to enable scientists to construct efficient RNA-guided gene drives not only in mosquitoes, but in many other species. In addition to altering populations of insects to prevent them from spreading disease (*Curtis, 1968*), this advance would represent an entirely new approach to ecological engineering with many potential applications relevant to human health, agriculture, biodiversity, and ecological science.

The first technical descriptions of endonuclease gene drives were provided by Austin Burt in his landmark proposal to engineer wild populations more than a decade ago (*Burt, 2003*). Any of the rapidly expanding number of laboratories with expertise in Cas9-mediated genome engineering could attempt to build a gene drive by substituting Cas9 for the homing endonucleases described in

his proposal. Indeed, the well-recognized potential for gene drives to combat vector-borne diseases such as malaria and dengue virtually ensures that this strategy will eventually be attempted in mosquitoes.

While considerable scholarship has been devoted to the question of how gene drives might be safely utilized in mosquitoes (*Scott et al., 2002*; *Touré et al., 2004*; *Benedict et al., 2008*; *Marshall, 2009*; *UNEP, 2010*; *Reeves et al., 2012*; *David et al., 2013*; *Alphey, 2014*), few if any studies have examined the potential ecological effects of gene drives in other species. After all, constructing a drive to spread a particular genomic alteration in a given species was simply not feasible with earlier genome editing methods. Disconcertingly, several published gene drive architectures could lead to extinction or other hazardous consequences if applied to sensitive species, demonstrating an urgent need for improved methods of controlling these elements. After consulting with experts in many fields as well as concerned environmental organizations, we are confident that the responsible development of RNA-guided gene drive technology is best served by full transparency and early engagement with the public.

Here we provide brief overviews of gene drives and Cas9-mediated genome engineering, detail the mechanistic reasons that RNA-guided gene drives are likely to be effective in many species, and outline probable capabilities and limitations. We further propose novel gene drive architectures that may substantially improve our control over gene drives and their effects, discuss possible applications, and suggest guidelines for the safe development and evaluation of this promising but as yet unrealized technology. A discussion of risk governance and regulation intended specifically for policymakers is published separately (*Oye et al., 2014*).

## Natural gene drives

In nature, certain genes 'drive' themselves through populations by increasing the odds that they will be inherited (*Burt and Trivers, 2009*). Examples include endonuclease genes that copy themselves into chromosomes lacking them (*Burt and Koufopanou, 2004*), segregation distorters that destroy competing chromosomes during meiosis (*Lyttle, 1991*), transposons that insert copies of themselves elsewhere in the genome (*Charlesworth and Langley, 1989*), Medea elements that eliminate competing siblings who do not inherit them (*Beeman et al., 1992*; *Chen*

*et al., 2007*), and heritable microbes such as *Wolbachia* (*Werren, 1997*).

### Endonuclease gene drives

Natural homing endonuclease genes exhibit drive by cutting the corresponding locus of chromosomes lacking them. This induces the cell to repair the break by copying the nuclease gene onto the damaged chromosome via homologous recombination (*Figure 1A*) (*Burt and Koufopanou, 2004*). The copying process is termed 'homing', while the endonuclease-containing cassette that is copied is referred to as a 'gene drive' or simply a 'drive'. Because copying causes the fraction of offspring that inherit the cassette to be greater than 1/2 (*Figure 1B*), these genes can drive through a population even if they reduce the reproductive fitness of the individual organisms that carry them. Over many generations, this self-sustaining process can theoretically allow a gene drive to spread from a small number of individuals until it is present in all members of a population.

### Engineered gene drives

To build an endonuclease gene drive, an endonuclease transgene must be inserted in place of a natural sequence that it can cut. If it can efficiently cut this sequence in organisms with one transgene and one natural locus, reliably induce the cell to copy the transgene, and avoid being too costly to the organism, it will spread through susceptible wild populations.

*Standard drives* spread genomic changes and associated traits through populations. Burt's original study proposed using them to drive the spread of other transgenes or to disrupt existing genes (*Figure 2A,B*) (*Burt, 2003*). The gene drive copying step can take place immediately upon fertilization (*Figure 2C*) or occur only in germline cells that are immediate precursors to sperm or eggs, leaving most of the organism's somatic cells with only one copy of the drive (*Figure 2D*). *Suppression drives* reduce the size of the targeted population. Austin Burt outlined an elegant strategy involving the use of gene drives to disrupt genes that cause infertility or lethality only when both copies are lost (*Burt, 2003*). These 'genetic load' drives would spread rapidly through minimally impaired heterozygotes when rare, and eventually cause the population to crash or even become extinct due to the accumulated load of recessive mutations. A second approach would mimic naturally occurring 'meiotic' or 'gametic' drives that bias the sex ratio (*Craig et al., 1960*; *Hickey and Craig, 1966*; *Hamilton, 1967*; *Newton et al., 1976*;

*Wood and Newton, 1991*). In this model, the Y chromosome (or its equivalent in other sex-determination systems) would encode an endonuclease that cuts and destroys the X chromosome during male meiosis, thereby ensuring that most viable sperm contain a Y chromosome (*Newton et al., 1976*; *Wood and Newton, 1991*; *Windbichler et al., 2007*, *2008*; *Galizi et al., 2014*). The progressively dwindling number of females will culminate in a population crash or extinction (*Craig et al., 1960*; *Lyttle, 1977*; *Burt, 2003*; *Schliekelman et al., 2005*; *Deredec et al., 2008*, *2011*; *North et al., 2013*; *Burt, 2014*).

Whether a standard gene drive will spread through a target population depends on molecular factors such as homing efficiency, fitness cost, and evolutionary stability (*Marshall and Hay, 2012*); only the rate of spread is determined by the mating dynamics, generation time, and other characteristics of the target population. In contrast, models suggest that the deleterious and complex effects of genetic load and sex-biasing suppression drives render them more sensitive to population-specific ecological variables such as density-dependent selection (*Burt, 2003*; *Schliekelman et al., 2005*; *Huang et al., 2007*; *Deredec et al., 2008*; *Marshall, 2009*; *Yahara et al., 2009*; *Deredec et al., 2011*; *Alphey and Bonsall, 2014*).

No engineered endonuclease gene drive capable of spreading through a wild population has yet been published. However, the Crisanti and Russell laboratories have constructed gene drives that can only spread through laboratory mosquito (*Windbichler et al., 2011*) and fruit fly (*Chan et al., 2011*; *Simoni et al., 2014*) populations that have been engineered to contain the endonuclease cut site. The Burt and Crisanti laboratories are attempting to build a male-biasing suppression drive using an endonuclease that serendipitously cuts a conserved sequence repeated hundreds of times in the X chromosome of the mosquito *Anopheles gambiae* (*Windbichler et al., 2007*, *2008*; *Galizi et al., 2014*). If successful, their work promises to substantially reduce the population of this important malaria vector.

All engineered gene drives based on homing endonucleases cut the natural recognition site of the relevant enzyme. Despite early hopes, it has proven difficult to engineer homing endonucleases to cleave new target sequences. Numerous laboratories have sought to accomplish this goal for well over a decade with only a few recent successes (*Chan et al., 2013a*; *Thyme et al., 2013*; *Takeuchi et al., 2014*). More recently, a team

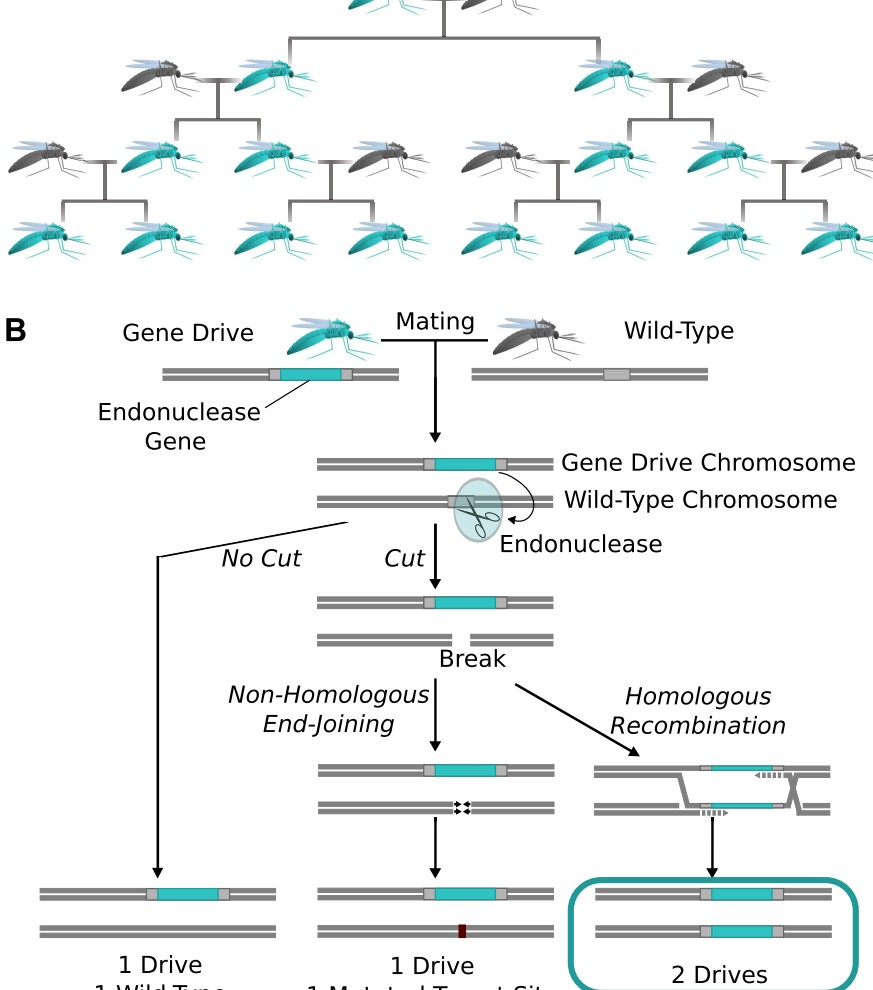

**A** Gene Drive    Wild-Type

**B** Gene Drive    Mating    Wild-Type

Endonuclease Gene

Gene Drive Chromosome
Wild-Type Chromosome
Endonuclease

No Cut    Cut

Break

Non-Homologous End-Joining    Homologous Recombination

1 Drive 1 Wild-Type    1 Drive 1 Mutated Target Site    2 Drives

**Figure 1**. The spread of endonuclease gene drives. (**A**) When an organism carrying an endonuclease gene drive (blue) mates with a wild-type organism (grey), the gene drive is preferentially inherited by all offspring. This can enable the drive to spread until it is present in all members of the population–even if it is mildly deleterious to the organism. (**B**) Endonuclease gene drives are preferentially inherited because the endonuclease cuts the homologous wild-type chromosome. When the cell repairs the break using homologous recombination, it must use the gene drive chromosome as a repair template, thereby copying the drive onto the wild-type chromosome. If the endonuclease fails to cut or the cell uses the competing non-homologous end-joining repair pathway, the drive is not copied, so efficient gene drives must reliably cut when homology-directed repair is most likely.

constructed new versions of the fruit fly gene drive using modular zinc-finger nucleases or TALENs in place of the homing endonuclease (*Simoni et al., 2014*), both of which can be engineered to cut new target sequences. While initially successful at

cutting and homing, both declined in effectiveness over time due to the evolutionary instability of the modular repeats inherent to those proteins.

These early attempts demonstrate that it is possible to build synthetic gene drives, but also emphasize the importance of cutting any desired gene and remaining stable during copying. The recent discovery of the RNA-guided Cas9 nuclease represents a possible solution.

### RNA-guided genome editing via the Cas9 nuclease

One straightforward method of genome editing relies on the same mechanism employed by endonuclease gene drives: cut the target gene and supply an edited version for the cell to use as a template when it fixes the damage. Most eukaryotic genome engineering over the past decade was accomplished using zinc-finger nucleases (*Urnov et al., 2005*) or TALENs (*Christian et al., 2010*), both of which are modular proteins that can be redesigned or evolved to target new sequences, albeit only by specialist laboratories (*Esvelt and Wang, 2013*). Genome editing was democratized by the discovery and adaptation of Cas9, an enzyme that can be programmed to cut target DNA sequences specified by a guiding RNA molecule (*Deltcheva et al., 2011*; *Jinek et al., 2012*; *Cho et al., 2013*; *Cong et al., 2013*; *Jinek et al., 2013*; *Mali et al., 2013c*).

Cas9 is a component of Type II CRISPR acquired immune systems in bacteria, which allow cells to 'remember' the sequences of previously encountered viral genomes and protect themselves by recognizing and cutting those sequences if encountered again. They accomplish this by incorporating DNA fragments into a memory element, transcribing it to produce RNAs with the same sequence, and directing Cas9 to cut any matching DNA sequences (*Deltcheva et al., 2011*). The only restriction is that Cas9 will only cut target 'protospacer' sequences that are flanked by a protospacer-adjacent motif (PAM) at the 3' end. The most commonly used Cas9 ortholog has a PAM with only two required bases (NGG) and therefore can cut protospacers found approximately every 8 base pairs (*Jinek et al., 2012*).

Remarkably, it is possible to direct Cas9 to cut a specific protospacer in the genome using only a single guide RNA (sgRNA) less than 100 base pairs in length (*Jinek et al., 2012*). This guide RNA must begin with a 17-20 base pair 'spacer' sequence identical to the targeted protospacer sequence in the genome (*Fu et al., 2014*).

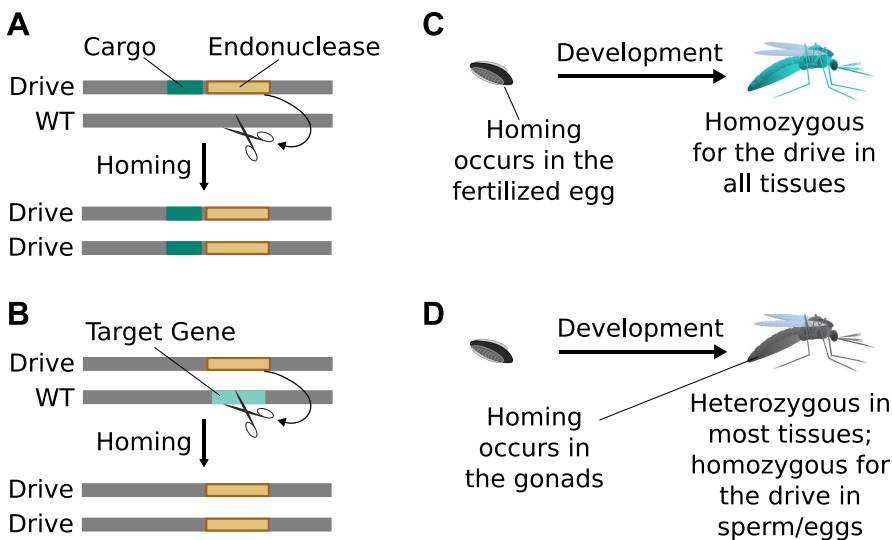

**Figure 2**. Consequences and timing of gene drive replication. (**A**) Gene drives can carry other genes with them as cargo. For example, a transgene that blocks malaria transmission could be driven through wild mosquito populations. There is no selection to maintain the function of a cargo gene. (**B**) Gene drives can disrupt or replace other genes. For example, a drive might replace a mosquito gene important for malaria transmission. Because it cannot spread without disrupting the target gene, this strategy is evolutionarily stable. (**C**) If homing occurs in the zygote or early embryo, all organisms that carry the drive will be homozygous in all of their tissues. (**D**) If homing occurs in the late germline cells that contribute to sperm or eggs, the offspring will remain heterozygous in most tissues and avoid the consequences of drive-induced disruptions.

The process of editing a target gene involves choosing protospacers within the gene, building one or more guide RNAs with matching spacers, and delivering Cas9, guide RNAs, and an edited repair template lacking those protospacers into the cell (**Figure 3**).

Cas9 is efficient enough to cut and edit multiple genes in a single experiment (*Li et al., 2013c*; *Wang et al., 2013a*). The enzyme is active in a wide variety of organisms and is also quite specific, cutting only protospacers that are nearly identical to the spacer sequence of the guide RNA (*Hsu et al., 2013*; *Mali et al., 2013a*; *Pattanayak et al., 2013*). Moreover, methods that allow Cas9 to bind but not cut enable the expression of target genes to be regulated by selectively recruiting regulatory proteins attached to Cas9 or the guide RNA (*Gilbert et al., 2013*; *Mali et al., 2013a*). All of these applications were developed within the last two years.

Because RNA-guided genome editing relies on exactly the same copying mechanism as endonuclease gene drives, it is reasonable to ask whether it might be possible to build gene drives based on Cas9. In principle, RNA-guided gene drives might be capable of spreading almost any genomic alteration that can be generated using Cas9 through sexually reproducing populations.

## Will RNA-guided gene drives enable us to edit the genomes of wild populations?

Although we cannot be certain until we try, current evidence suggests that RNA-guided gene drives will function in some and possibly most sexually reproducing species. Learning how to insert a drive into the germline and optimize its function in each new species will likely require months to years depending on generation length, with subsequent drives in the same species taking less time. Because inserting the drive into the germline with Cas9 involves the same molecular copying process as the drive itself will utilize, successful insertion may produce a working if not particularly efficient RNA-guided gene drive. But if population-level engineering is to become a reality, all molecular factors relevant to homing – cutting, specificity, copying, and evolutionary robustness – must be considered. Below, we provide a detailed technical analysis of the extent to which Cas9 can address each of these challenges. Capabilities, limitations, control strategies, and possible applications are discussed in subsequent sections.

### *Cutting*

The first requirement for every endonuclease gene drive is to cut the target sequence. Incomplete cutting was a problem for the homing endonuclease drive constructed in transgenic mosquitoes (72% cutting) and also for the homing endonuclease, zinc-finger nuclease, and TALEN drives in fruit flies (37%, 86%, and 70% cutting) (*Windbichler et al., 2011*; *Chan et al., 2013b*; *Simoni et al., 2014*). The simplest way to increase cutting is to target multiple adjacent sequences. However, this is impractical for homing endonucleases and quite difficult for zinc-finger nucleases and TALENs, as each additional sequence requires a new nuclease protein to be engineered or evolved and then co-expressed.

In contrast, the RNA-guided Cas9 nuclease can be readily directed to cleave additional sequences by expressing additional guide RNAs

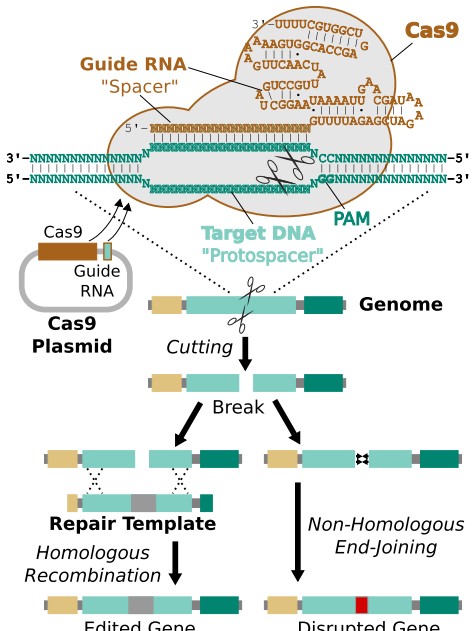

**Figure 3**. RNA-guided genome editing via Cas9. The Cas9 nuclease protein and guide RNA must first be delivered into the target cell. This is often accomplished by transfecting DNA expression plasmids, but delivering RNA is also effective. The guide RNA directs Cas9 to bind target DNA 'protospacer' sequences that match the 'spacer' sequence within the guide RNA. Protospacers must be flanked by an appropriate protospacer-adjacent motif (PAM), which is NGG for the most commonly used Cas9 protein (*Jinek et al., 2012*). If the spacer and protospacer are identical or have only a few mismatches at the 5' end of the spacer, Cas9 will cut both strands of DNA, creating a blunt-ended double-strand break. If supplied with a repair template containing the desired changes and homology to the sequences on either side of the break, the cell may use homologous recombination to repair the break by incorporating the repair template into the chromosome. Otherwise, the break will be repaired by non-homologous end-joining, resulting in gene disruption. Cas9 cutting is efficient enough to alter both chromosomes at the same time and/or to edit multiple genes at once (*Li et al., 2013*; *Wang et al., 2013a*). If the cell being edited is a germline cell that gives rise to eggs or sperm, the changes can be inherited by future generations.

(*Figure 4*). The sequences of these additional guide RNAs can be altered so as to avoid creating unstable repeats within the drive cassette (*Nishimasu et al., 2014*; *Simoni et al., 2014*). Including more guides has been demonstrated to improve upon already high rates of cutting. For example, fruit flies expressing both Cas9 and guide RNAs in their germline exhibited target cutting rates exceeding 85–99% in males for four

out of six tested guide RNAs (*Kondo and Ueda, 2013*). The two least effective guide RNAs individually cut at rates exceeding 12% and 56%, but exhibited cutting rates above 91% when combined. Using more than two guide RNAs should further enhance cutting. The notable success of Cas9-based genome engineering in many different species, including studies that targeted every gene in the genome (*Shalem et al., 2014*; *Wang et al., 2014*), demonstrates that most sequences can be efficiently targeted independent of species and cell type. Thus, RNA-guided gene drives should be capable of efficiently cutting any given gene.

## Specificity

Because cutting other sites in the genome may seriously compromise the fitness of the organism, the second requirement is to avoid cutting non-targeted sequences.

While several studies have reported that Cas9 is prone to cutting off-target sequences that are closely related to the target (*Fu et al., 2013*; *Hsu et al., 2013*; *Mali et al., 2013a*; *Pattanayak et al., 2013*), more recent developments and strategies designed to improve specificity (*Mali et al., 2013a*; *Fu et al., 2014*; *Guilinger et al., 2014*; *Tsai et al., 2014*) have demonstrated that the off-target rate can be reduced to nearly undetectable levels (*Figure 4*). Notably, Cas9 does not appear to represent a noticeable fitness burden when expressed at a moderate level in fruit flies with or without guide RNAs (*Kondo and Ueda, 2013*). Organisms with larger genomes may require more careful target site selection due to the increased number of potential off-target sequences present.

## Copying

The third and most challenging requirement involves ensuring that the cut sequence is repaired using homologous recombination (HR) to copy the drive rather than the competing non-homologous end-joining (NHEJ) pathway (*Figure 4*). HR rates are known to vary across cell types (*Mali et al., 2013c*), developmental stages (*Fiorenza et al., 2001*; *Preston et al., 2006*), species (*Chan et al., 2013b*), and the phase of the cell cycle (*Saleh-Gohari and Helleday, 2004*). For example, the endonuclease gene drive in mosquitoes was correctly copied following ~97% of cuts (*Windbichler et al., 2011*), while a similar drive in fruit flies was initially copied only 2% of the time (*Chan et al., 2011*) and never rose above 78% even with extensive combinatorial optimization of promoter and 3' untranslated region. This difference is presumably

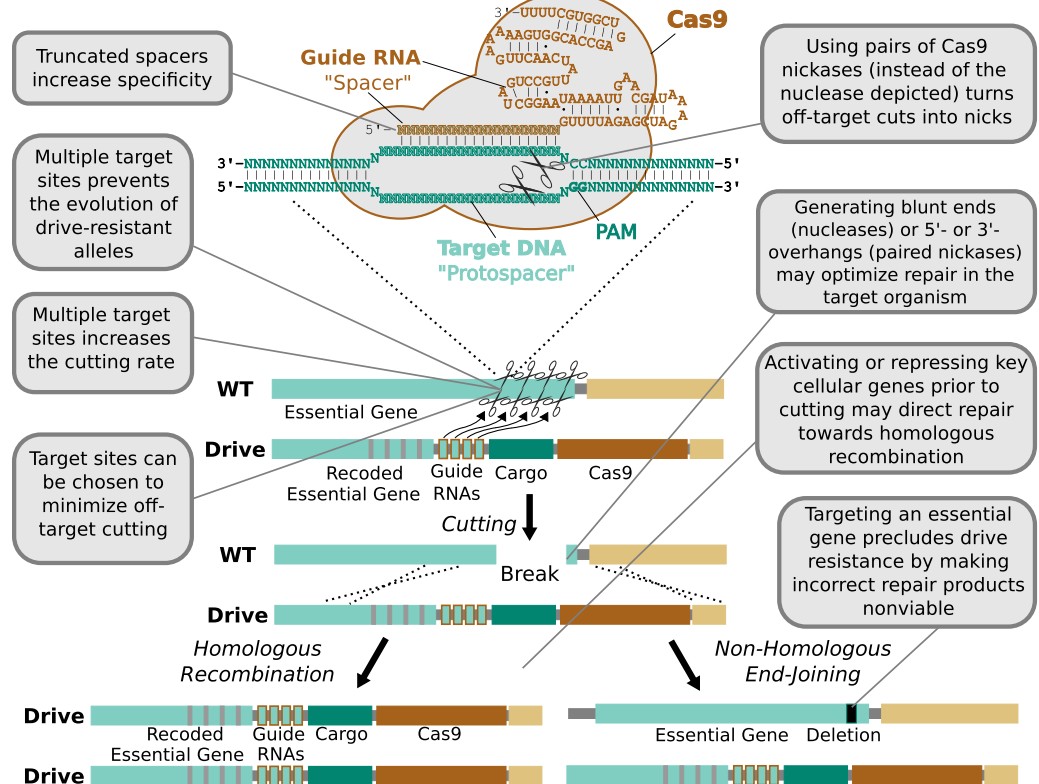

**Figure 4**. Technical advantages of RNA-guided gene drives. Clockwise from lower left: The targeting flexibility of Cas9 permits the exclusive selection of target sequences with few potential off-targets in the genome. Targeting multiple sites increases the cutting frequency and hinders the evolution of drive resistant alleles, which must accumulate mutations at all of the sites. The Cas9 nuclease is can be quite specific in the sequences that it targets; fruit flies do not exhibit notable fertility or fitness defects resulting from off-target cutting when both Cas9 nuclease and guide RNAs are expressed in the germline (**Kondo and Ueda, 2013**). Choosing target sites with few or no close relatives in the genome, using truncated guide RNAs (**Fu et al., 2014**), employing paired Cas9 nickases (**Mali et al., 2013a**) instead of nucleases, or utilizing Cas9-FokI fusion proteins (**Guilinger et al., 2014**; **Tsai et al., 2014**) can further increase specificity. Several of these strategies can reduce the off-target mutation rate to borderline undetectable levels (**Fu et al., 2014**; **Guilinger et al., 2014**; **Tsai et al., 2014**). The frequency at which the drive is correctly copied might be increased by using Cas9 as a transcriptional regulator to activate HR genes and repress NHEJ genes (**Gilbert et al., 2013**; **Mali et al., 2013a**) (**Figure 4—figure supplement 1**). By choosing target sites within an essential gene, any non-homologous end-joining event that deletes all of the target sites will cause lethality rather than creating a drive-resistant allele, further increasing the evolutionary robustness of the RNA-guided gene drive. Other options include using distinct promoters and guide RNAs to avoid repetitiveness and increase stability (**Figure 4—figure supplement 2**) or employing newly characterized, engineered, or evolved Cas9 variants with improved properties (**Esvelt et al., 2011**; **Mali et al., 2013b**). These optimization strategies have also been summarized in tabular form with additional details (**Figure 4—figure supplement 3**).

The following figure supplements are available for figure 4:

**Figure supplement 1**. Enhancing drive copying by regulating endogenous genes.

**Figure supplement 2**. Repetitiveness and evolutionary stability of multiple guide RNAs.

**Figure supplement 3**. Table of known technological advances that might be adapted to optimize gene drive efficiency.

due to a lower rate of HR in fruit fly spermatocytes relative to mosquitoes (*Chan et al., 2013b*). Ideally, drives should be activated only in germline cells at developmental stages with a high rate of HR, but this may be challenging in many species.

Copying efficiencies may also depend on whether the cut produces 5′-overhangs, 3′-overhangs, or blunt ends (*Kuhar et al., 2014*). Because Cas9 nickases can generate either overhang type while Cas9 nucleases produce blunt ends, the enzyme can be adapted to the needs of the cell type and organism.

The ability to regulate gene expression with Cas9 might be used to temporarily increase the rate of homologous recombination while the drive is active (*Figure 4*). For instance, the Cas9 nuclease involved in cutting might simultaneously repress (*Gilbert et al., 2013*) genes involved in NHEJ and therefore increase the frequency of HR (*Bozas et al., 2009*) if supplied with a shortened guide RNA that directs it to bind and block transcription but not cut (*Bikard et al., 2013*; *Sternberg et al., 2014*). Alternatively, an orthogonal nuclease-null Cas9 protein (*Esvelt et al., 2013*) encoded within the drive cassette could repress NHEJ genes and activate HR genes before activating the Cas9 nuclease. Lastly, Cas9 might be used directly recruit key HR-directing proteins to the cut sites, potentially biasing repair towards that pathway.

### Evolutionary stability

Even a perfectly efficient endonuclease gene drive is vulnerable to the evolution of drive resistance in the population. Whenever a cut is repaired using the NHEJ pathway, the result is typically a drive-resistant allele with insertions or deletions in the target sequence that prevent it from being cut by the endonuclease. Natural sequence polymorphisms in the population could also prevent cutting. These alleles will typically increase in abundance and eventually eliminate the drive because most drives – like most transgenes – are likely to slightly reduce the fitness of the organism. A second path to gene drive resistance would involve the target organism evolving a method of specifically inhibiting the drive endonuclease.

The best defense against previously existing or recently evolved drive-resistant alleles is to target multiple sites. Because mutations in target sites are evolutionarily favored only when they survive confrontation with the gene drive, using many target sites can render it statistically improbable for any one allele to survive

long enough to accumulate mutations at all of the sites so long as cutting rates are high (*Burt, 2003*). However, very large populations – such as those of some insects – might require unfeasibly large numbers of guide RNAs to prevent resistance. In these cases it may be necessary to release several successive gene drives, each targeting multiple sites, to overcome resistant alleles as they emerge. From an evolutionary perspective, the ability to preclude resistance by targeting multiple sites is the single greatest advantage of RNA-guided gene drives.

We propose to extend this strategy by preferentially targeting multiple sites within the 3′ ends of genes important for fitness such that any repair event that deletes all of the target sites creates a deleterious allele that cannot compete with the spread of the drive (*Figure 4*). Whenever the drive is copied, the cut gene is replaced with a recoded version flanked by the other components of the drive. Recent work has demonstrated that most genes can be substantially recoded with little effect on organism fitness (*Lajoie et al., 2013*); the 3′ untranslated region might be replaced with an equivalent sequence from a related gene. Because there would be no homology between the recoded cut site and the drive components, the drive cassette would always be copied as a unit.

Relative to drive-resistant alleles, inhibitors of Cas9 are less likely to arise given the historical absence of RNA-guided nucleases from eukaryotes. Any inhibitors that do evolve would presumably target a particular Cas9 protein or guide RNA used in an earlier drive and could be evaded by building future drives using a Cas9 ortholog with a different guide RNA (*Esvelt et al., 2013*; *Fonfara et al., 2013*). Alternatively and least likely, organisms might evolve higher RNase activity to preferentially degrade all guide RNAs; this may be difficult to accomplish without harming overall fitness.

A final evolutionary concern relates to the stability of the gene drive cassette itself. The zinc-finger nuclease and TALEN-based gene drives in fruit flies suffered from recombination between repetitive sequences: only 75% and 40% of each respective drive was sufficiently intact after one copying event to catalyze a second round of copying. Because RNA-guided gene drives will not include such highly repetitive elements, they are likely to be more stable (*Figure 4*).

### Development time

RNA-guided genome editing is advancing at a historically unprecedented pace. Because it is

now much easier to make transgenic organisms and therefore candidate gene drives, the design-build-test cycle for gene drives will often be limited only by the generation time of the organism in the laboratory. Moreover, many advances from genome engineering can be directly applied to RNA-guided gene drives. For example, all of the methods of increasing Cas9 specificity described above were developed for RNA-guided genome editing in the past 2 years. Future methods of increasing the rate of HR relative to NHEJ would be useful for both technologies. These factors suggest that scientists will enjoy an increasing number of tools well-suited to rapidly building and testing gene drives in addition to those we describe above.

None of this is to gloss over the many practical difficulties that are likely to arise when constructing a particular gene drive in a given species. Early success is as unlikely as ever when engineering complex biological systems. But if half a dozen or even a dozen design-build-test cycles are sufficient to produce moderately efficient gene drives, many molecular biology laboratories around the world will soon be capable of engineering wild populations.

## Gene drive limitations

Given their potentially widespread availability, it will be essential to develop a comprehensive understanding of the fundamental limitations of genetic drive systems.

First and most important, gene drives require many generations to spread through populations. Once transgenic organisms bearing the gene drive are constructed in the laboratory, they must be released into the wild to mate with wild-type individuals in order to begin the process of spreading the drive through the wild population. The total time required to spread to all members depends on the number of drive-carrying individuals that are released, the generation time of the organism, the efficiency of homing, the impact of the drive on individual fitness, and the dynamics of mating and gene flow in the population, but in general it will take several dozen generations (*Burt, 2003*; *Huang et al., 2007*; *Deredec et al., 2008*; *Marshall, 2009*; *Yahara et al., 2009*; *Deredec et al., 2011*). Thus, drives will spread very quickly in fast-reproducing species but only slowly in long-lived organisms.

Second, gene drives cannot affect species that exclusively practice asexual reproduction through clonal division or self-fertilization. This

> ## Box 1. Could gene drives alter human populations?
>
> Not unless we wait for many centuries. Even in a hypothetical future in which human germline editing was considered safe and ethical, a driven alteration would be only four times as abundant as a non-driven alteration a full century after the birth of an edited human. This assumes future generations would not elect to remove the drive. Whole-genome sequencing - a technology that is available in many modern hospitals and is widely expected to be ubiquitous in the near future - is quite capable of detecting the presence of any gene drive if we decide to look.

category includes all viruses and bacteria as well as most unicellular organisms. Highly efficient standard drives might be able to slowly spread through populations that employ a mix of sexual and asexual reproduction, such as certain plants, but drives intended to suppress the population would presumably force target organisms to reproduce asexually in order to avoid suppression.

Third, drive-mediated genome alterations are not permanent on an evolutionary timescale. While gene drives can spread traits through populations even if they are costly to each individual organism, harmful traits will eventually be outcompeted by more fit alleles after the drive has gone to fixation. Highly deleterious traits may be eliminated even more quickly, with non-functional versions appearing in large numbers even before the drive and its cargo can spread to all members of the population. Even when the trait is perfectly linked to the drive mechanism, the selection pressure favoring the continued function of Cas9 and the guide RNAs will relax once the drive reaches fixation. Maintaining deleterious traits within a population indefinitely is likely to require scheduled releases of new RNA-guided gene drives to periodically overwrite the broken versions in the environment.

Fourth, our current knowledge of the risk management (*Scott et al., 2002*; *Touré et al., 2004*; *UNEP, 2010*; *McGraw and O'Neill, 2013*;

*Alphey, 2014*) and containment (*Benedict et al., 2008*; *Marshall, 2009*) issues associated with gene drives is largely due to the efforts of researchers focused on mosquito-borne illnesses. Frameworks for evaluating ecological consequences are similarly focused on mosquitoes (*David et al., 2013*) and the few other organisms for which alternative genetic biocontrol methods have been considered (*Dana et al., 2014*). While these examples provide an invaluable starting point for investigations of RNA-guided gene drives targeting other organisms, studies examining the particular drive, population, and associated ecosystem in question will be needed.

## Safeguards and control strategies

Given the potential for gene drives to alter entire wild populations and therefore ecosystems, the development of this technology must include robust safeguards and methods of control (*Oye et al., 2014*). Whereas existing gene drive proposals focus on adding genes (*Ito et al., 2002*), disrupting existing genes (*Burt, 2003*), or suppressing populations, RNA-guided gene drives will also be capable of replacing existing sequences with altered versions that have been recoded to remove the sites targeted by the drive (*Figure 3*). We hypothesize that the unique

---

> # Box 2. Could gene drives alter domesticated animals or crops?
>
> In theory they could, though probably not without the permission of the farmers, scientists and breeders who typically monitor reproduction. For example, genetic records and artificial insemination are so common among cattle and other domesticated animals that it would be exceedingly difficult for a gene drive to spread through any of these species. Seed farms play a similar role for crops. Long generation times will represent a further barrier for many domesticated species. In general, our expectation is that beneficial applications are more likely to involve the alteration of weeds and insect pest populations rather than the crops themselves.

---

ability of RNA-guided gene drives to target any gene may allow them to control the effects of other gene drives or transgenes.

## Reversibility

RNA-guided gene drives could reverse genome alterations that have already spread through populations. Suppose a given gene drive causes unexpected side-effects or is released without public consent. A 'reversal' drive released later could overwrite one or all genomic changes spread by the first drive (*Figure 5A*). The new sequence spread by the reversal drive must also be recoded relative to the original to keep the first drive from cutting it, but any amino acid changes introduced by the first drive could be undone. If necessary, a third drive could restore the exact wild-type sequence, leaving only the guide RNAs and the gene encoding Cas9 as signatures of past editing (*Figure 5B*).

The ability to update or reverse genomic alterations at the speed of a drive, not just a drive-resistant allele, represents an extremely important safety feature. Reversal drives could also remove conventionally inserted transgenes that entered wild populations by cross-breeding or natural mutations that spread in response to human-induced selective pressures. However, it is important to note that even if a reversal drive were to reach all members of the population, any ecological changes caused in the interim would not necessarily be reversed.

### Immunization

RNA-guided gene drives could be used to block the spread of other gene drives. For example, an 'immunizing' drive could prevent a specific unwanted drive from being copied by recoding sequences targeted by the unwanted drive (*Figure 5A*). This could be done preemptively or reactively and would spread on a timescale comparable to that of the unwanted drive. A combined 'immunizing reversal' drive might spread through both wild-type individuals and those affected by an earlier gene drive, converting both types to a recoded version that could not be invaded by the unwanted drive (*Figure 5B*). This may represent the fastest method of neutralizing an already-released drive. As with a standard reversal drive, any ecological changes caused in the interim would not necessarily be reversed.

### Precisely targeting subpopulations

RNA-guided gene drives might be confined to a single genetically distinct target species

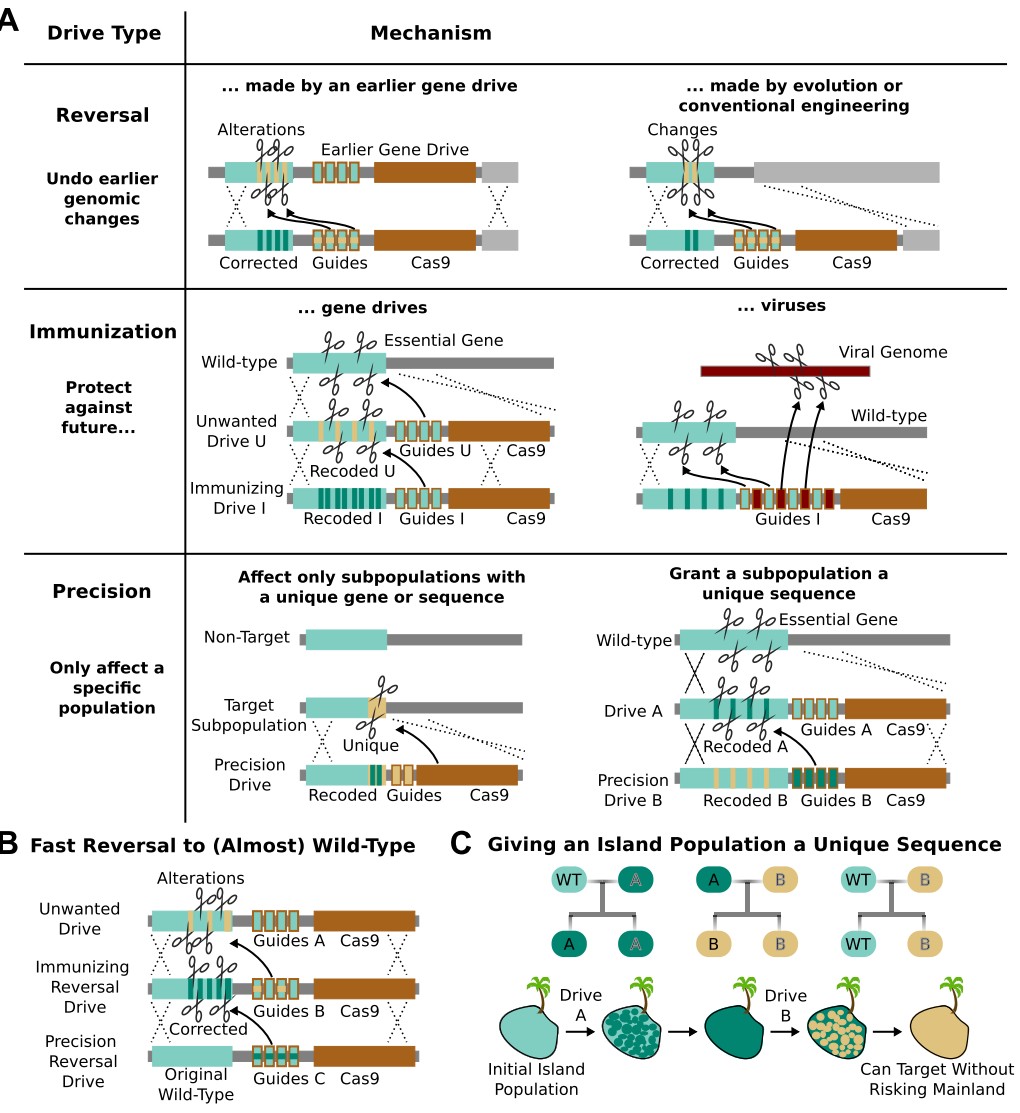

**Figure 5**. Methods of reversing, preventing, and controlling the spread and effects of gene drives. (**A**) *Reversal drives* could correct or reverse genomic alterations made by an earlier drive with unexpected side effects. They might also be used to reverse conventionally engineered or evolved changes. *Immunization drives* could prevent other gene drives from affecting a specific population or provide a population with resistance to DNA viruses. *Precision drives* could exclusively spread through a subpopulation with a unique gene or sequence. (**B**) Together, these can quickly halt an unwanted drive and eventually restore the sequence to the original wild-type save for the residual Cas9 and guide RNAs. (**C**) Any population with limited gene flow can be given a unique sequence by releasing drives **A** and **B** in quick succession. So long as drive **A** does not escape into other populations before it is completely replaced by drive **B**, subsequent precision drives can target population **B** without risking spread into other populations.

or even a subpopulation by targeting unique genes or sequence polymorphisms. Because these 'precision drives' will only cut the unique sequence, they will not be able to spread through non-target populations as long as that sequence is sufficiently distinct. We estimate that either the PAM or at least five base pairs of the spacer must differ within each target site in order to prevent the guide RNAs in the drive from evolving to recognize

the equivalent non-target sequence (*Fu et al., 2013*; *Hsu et al., 2013*; *Mali et al., 2013a*; *Pattanayak et al., 2013*).

Populations that are not genetically distinct but experience only intermittent gene flow, such as those on islands, might be given a unique sequence permitting them to be specifically targeted by precision drives later on. For example, releasing Drive A into the island population

would recode a target gene, but exhibit no other effect (*Figure 5A*). Releasing Drive B, a precision drive which would exclusively spread through Drive A but not the wild-type allele, would similarly replace Drive A with a unique sequence. So long as Drive A does not escape the island before being replaced, the unique sequence in the island population would allow it to be targeted with future precision drives that could not spread through mainland populations (*Figure 5C*).

### Limiting population suppression

Population suppression may be one of the most powerful applications of gene drives. The previously described genetic load and sex-biasing drives (*Burt, 2003*) could potentially lead to extinction (*Deredec et al., 2008*, *2011*). While this outcome may be necessary to achieve compelling goals such as the eradication of malaria, other situations may call for more refined methods. Here we outline a handful of alternative architectures that would provide greater control over the extent of population suppression.

Chemical approaches to population control might utilize 'sensitizing drives' to render target organisms vulnerable to a particular molecule using one of three strategies (*Figure 6*). First, a sensitizing drive might reverse known mutations that confer resistance to existing pesticides or herbicides. Second, it might carry a prodrug-converting enzyme (*Schellmann et al., 2010*) that would render a prodrug molecule toxic to organisms that express it. Third, it could swap a conserved gene for a version that is strongly inhibited by a particular small molecule. Because sensitizing drives would have no effect in the absence of their associated molecule – and in some cases vice versa – they could grant very fine control over the geography and extent of population suppression with minimal ecological risk.

Temporal approaches to controlling populations would deliberately limit the lifetime of a suppression drive by rendering its effects evolutionarily unstable (*Figure 6*). For example, a male-determining or female-specific sterility gene carried by a standard drive on an autosome would suppress the target population, but the effect would be short-lived because any drive that acquired a loss-of-function mutation in the cargo gene would be strongly favored by natural selection due to its ability to produce fertile female offspring. Notably, turning existing female-specific sterility lines (*Fu et al., 2010*; *Labbe et al., 2012*; *Alphey, 2014*) into unstable drives may increase

their effectiveness. Periodically releasing organisms carrying new unstable drives that are capable of replacing earlier broken versions could extend the suppression effect.

Genetic approaches to population control might initiate suppression only when two distinct 'interacting drives' encounter one another (*Figure 6*). For example, a cross between standard drives A and B might produce sterile females and fertile males that pass on the 'sterile-daughter' trait when crossed with females of any type. Scattering A- and B-carrying individuals throughout an existing population would produce many tiny pockets dominated by either A or B and very few organisms in between due to the infertility of AB females. Because each drive would spread from a small number of initially released individuals scattered over a wide area, this strategy may be capable of large-scale population suppression, but its effectiveness and resolution will depend on the average distance between released A and B individuals. Further suppressing the residual A and B populations could be accomplished by releasing only members of the opposite drive type. Modeling studies will be needed to determine whether this possibility is feasible for different species. Interestingly, the use of this drive type would effectively induce speciation in the affected population.

Finally, immunizing drives might protect specific subpopulations from the effects of full-scale suppression drives released elsewhere (*Figure 6*). Assuming some degree of gene flow, the immunized population will eventually replace the suppressed population, though this might be delayed if crossing the two drives generates a sterile-daughter effect as described above. Due to the comparatively uncontrolled spread of both drive types through the wild-type population, this method would only be suited to large geographic areas or subpopulations with limited gene flow. For example, immunization might be used to protect the native population of a species while suppressing or eradicating populations on other continents.

## Applications of RNA-guided gene drives

RNA-guided gene drives have the potential to merge the fields of genomic and ecological engineering. They may enable us to address numerous problems in global health, agriculture, sustainability, ecological science, and many other areas (*Figure 7*). Of these opportunities, perhaps the most compelling involve curtailing the spread

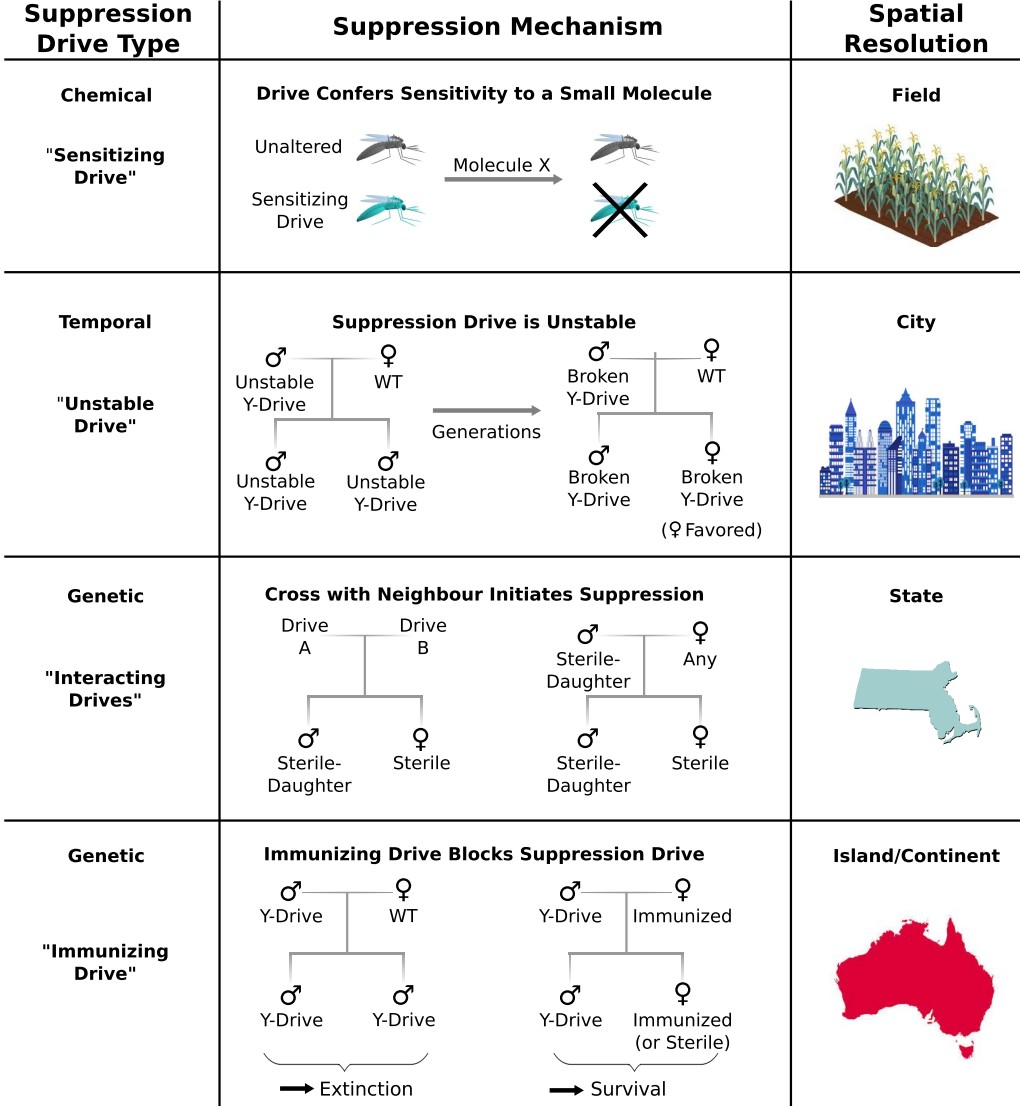

| Suppression Drive Type | Suppression Mechanism | Spatial Resolution |
|---|---|---|
| **Chemical**<br><br>**"Sensitizing Drive"** | **Drive Confers Sensitivity to a Small Molecule** | **Field** |
| **Temporal**<br><br>**"Unstable Drive"** | **Suppression Drive is Unstable** | **City** |
| **Genetic**<br><br>**"Interacting Drives"** | **Cross with Neighbour Initiates Suppression** | **State** |
| **Genetic**<br><br>**"Immunizing Drive"** | **Immunizing Drive Blocks Suppression Drive** | **Island/Continent** |

**Figure 6**. Controlling population suppression. Previously proposed genetic load and meiotic suppression drives spread without limit and may incur a substantial risk of extinction. Alternative gene drive types might be used to grant finer control over the extent of suppression. 'Sensitization drives' would be harmless save for conferring vulnerability to a particular chemical, which could then be used as a population-specific pesticide. Evolutionarily 'unstable drives' would place a limit on the average number of drive copying events and thus the extent of population suppression. 'Interacting drives' would initiate suppression only upon encountering a specific genetic signature in the population, in this case a different gene drive. The combination would create a sterile-daughter effect (***Figure 6—figure supplement 1***) capable of continuing suppression for several generations. Finally, an immunizing drive could protect a subpopulation from a full genetic load or male-biasing suppression drive employed elsewhere. Interacting drive and immunizing drive approaches would be effective on very large populations spread across substantial geographic areas (***Figure 6—figure supplement 2***) while suffering from correspondingly reduced geographic resolution and greater ecological risk (***Figure 6—figure supplement 3***). Resolutions are approximations only and will vary with the specific drive utilized in each class.

The following figure supplements are available for figure 6:

**Figure supplement 1**. Sample interacting drives that produce a sterile-daughter effect.

*Figure 6. Continued on next page*

*Figure 6. Continued*

**Figure supplement 2**. An extreme example of ecological management: the use of suppression and immunizing drives to control rat populations worldwide.

**Figure supplement 3**. Characteristics of population suppression drives.

## Box 3. What types of genes can be edited using gene drives?

Genes can be edited reliably if they are important to fitness. This is because NHEJ events that create drive-resistant alleles by deleting all the protospacer cut sites will only be harmful if they disrupt the function of important genes. NHEJ events in unimportant genes and sequences will produce drive-resistant alleles lacking the targeted sites. These will spread and interfere with propagation of the drive. As a result, unimportant genes can be reliably disrupted or deleted but not edited.

Genes that are carried as cargo will not be evolutionarily stable unless they directly contribute to the efficient function of the drive. This limits opportunities to spread transgenes unrelated to drive function, although periodically releasing new drives that overwrite earlier broken versions could potentially maintain functional cargo genes in a large fraction of the population.

of vector-borne infectious diseases, controlling agricultural pests, and reducing populations of environmentally and economically destructive invasive species.

### Eradicating insect-borne diseases

The human toll inflicted by infectious diseases spread by insects is staggering. Malaria alone kills over 650,000 people each year, most of them children, while afflicting 200 million more with debilitating fevers (*WHO, 2013*). Dengue, yellow fever, chikungunya, trypanosomiasis, leishmaniasis, Chagas, and Lyme disease are also spread by insects. These afflictions could potentially be controlled or even eradicated by altering vector

species to block transmission. Several laboratories have identified candidate gene disruptions or transgenes that interfere with the transmission of malaria (*Ito et al., 2002*; *Dong et al., 2011*; *Isaacs et al., 2012*) and other well-studied diseases (*Franz et al., 2006*). Depending on their effectiveness, these alterations may or may not allow the disease to be eradicated before the pathogen evolves resistance. Alternatively, the relevant vector species might be suppressed or eliminated using RNA-guided gene drives, then potentially reintroduced from sheltered laboratory or island populations once disease eradication is complete. In the case of malaria, gene drive strategies may represent particularly effective solutions to the emerging problem of mosquito vectors with an evolved preference to bite and rest outdoors, traits that render them resistant to current control strategies based on indoor insecticide spraying and bednets.

### Agricultural safety and sustainability

The evolution of resistance to pesticides and herbicides is a major problem for agriculture. It has been assumed that resistant populations will remain resistant unless the relevant alleles impose a substantial fitness cost in the absence of pesticide or herbicide. We propose that RNA-guided sensitizing drives might replace resistant alleles with their ancestral equivalents to restore vulnerability. For example, sensitizing drives could potentially reverse the mutations allowing the western corn rootworm to resist Bt toxins (*Gassmann et al., 2014*) or horseweed and pigweed to resist the herbicide glyphosate (*Gaines et al., 2010*; *Ge, 2010*), which is currently essential to more sustainable no-till agriculture. Because these three organisms undergo one generation per year, comparatively large numbers of drive-bearing individuals must be released to quickly exert an effect, but fewer than are already released to control pests using the sterile-insect technique (*Gould and Schliekelman, 2004*; *Dyck et al., 2005*). Releases would need to occur in local areas not treated with pesticide or herbicide, which would quickly become reservoirs of sensitizing drives that could spread into adjacent

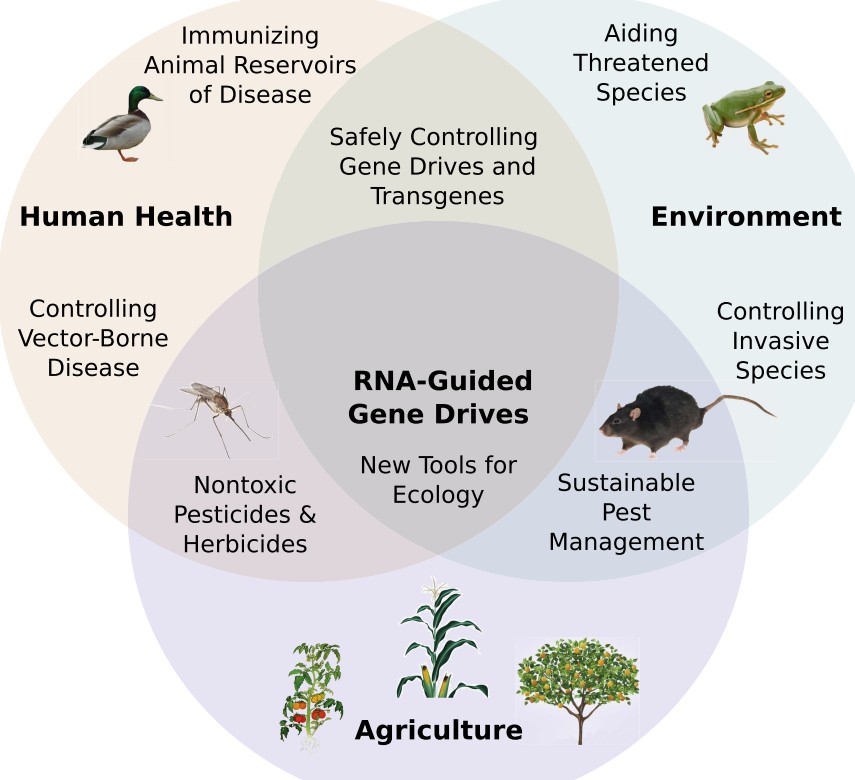

**Figure 7**. Potential applications of RNA-guided gene drives. Clockwise from left. Disease vectors such as malarial mosquitoes might be engineered to resist pathogen acquisition or eliminated with a suppression drive. Wild populations that serve as reservoirs for human viruses could be immunized using Cas9, RNAi machinery, or elite controller antibodies carried by a gene drive. Reversal and immunization drives could help ensure that all transgenes are safe and controlled. Drives might quickly spread protective genes through threatened or soon-to-be-threatened species such as amphibians facing the expansion of chytrid fungus (*Rosenblum et al., 2010*). Invasive species might be locally controlled or eradicated without directly affecting others. Sensitizing drives could improve the sustainability and safety of pesticides and herbicides. Gene drives could test ecological hypotheses concerning gene flow, sex ratios, speciation, and evolution. Technical requirements for these applications vary with the drive type required (*Figure 7—figure supplement 1*).

The following figure supplement is available for figure 7:

**Figure supplement 1**. Technical limitations of different gene drive architectures with implications for various applications.

fields. Periodically releasing new drives could potentially allow any given pesticide or herbicide to be utilized indefinitely. Modeling experiments will be needed to evaluate feasibility for different target species.

A second form of sensitizing drive could potentially render pest populations vulnerable to molecules that never previously affected them. For example, a gene important to fitness might be replaced with a version from another species or laboratory isolate whose function is sensitive to a particular compound. In principle, this approach could eventually lead to the development and use of safer and more species-specific pesticides and herbicides.

### Controlling invasive species

One of the most environmentally damaging consequences of global economic activity is the introduction of invasive species, which often cause ecological disruption or even the extinction of native species. Isolated ecosystems such as those on small islands are especially vulnerable. RNA-guided suppression drives might be used to promote biodiversity by controlling or even eradicating invasive species from islands or possibly entire continents. The economics of invasive species control are also compelling: the top ten invasives in the United States cause an estimated $42 billion in damages every year (*Pimentel et al., 2005*). Black and brown rats alone cause $19 billion in damages and may be responsible for more extinctions than any other nonhuman species.

Deploying RNA-guided suppression drives against invasive species will incur two primary risks related to undesired spread. First, rare mating events may allow the drive to affect closely related species. Using precision drives to target sequences unique to the invasive species could mitigate or eliminate this problem. Second, the suppression drive might spread from the invasive population back into the native habitat, perhaps even through intentional human action.

Native populations might be protected using an immunizing drive, but doing so would risk transferring immunity back into invasive populations. Instead, we might grant the invasive population a unique sequence with a standard drive (*Figure 5C*), verify that these changes have not spread to the native population, and only then release a suppression drive targeting the recoded sequences while holding an immunizing drive in reserve. Another approach might utilize a sensitizing drive to render all populations newly

vulnerable to a specific compound, which could then be used as a pesticide for the local control of invasive populations. All of these possibilities will require modeling and experimentation to establish safety and feasibility before use.

Most importantly, all decisions involving the use of suppression drives must involve extensive deliberations including but not limited to ecologists and citizens of potentially affected communities.

## Development and release precautions

Because any consequences of releasing RNA-guided gene drives into the environment would be shared by the local if not global community, research involving gene drives capable of spreading through wild-type populations should occur only after a careful and fully transparent review process. However, basic research into gene drives and methods of controlling their effects can proceed without risking this type of spread so long as appropriate ecological or molecular containment strategies are employed (*Figure 8*).

A great deal of information on probable ecological outcomes can be obtained without testing or even building replication-competent gene drives. For example, early studies might examine possible ecological effects by performing contained field trials with organisms that have been engineered to contain the desired change but do not possess a functional drive to spread it. Because they do involve transgenic organisms, these experiments are not completely without risk, but such transgenes are unlikely to spread in the absence of a drive.

We recommend that all laboratories seeking to build standard gene drives capable of spreading through wild populations simultaneously create reversal drives able to restore the original phenotype. Similarly, suppression drives should be constructed in tandem with a corresponding immunizing drive. These precautions would allow the effects of an accidental release to be swiftly if partially counteracted. The prevalence of gene drives in the environment could in principle be monitored by targeted amplification or metagenomic sequencing of environmental samples. Further investigation of possible monitoring strategies will be needed.

## Transparency, public discussion, and evaluation

Technologies with the potential to significantly influence the lives of the general public demand

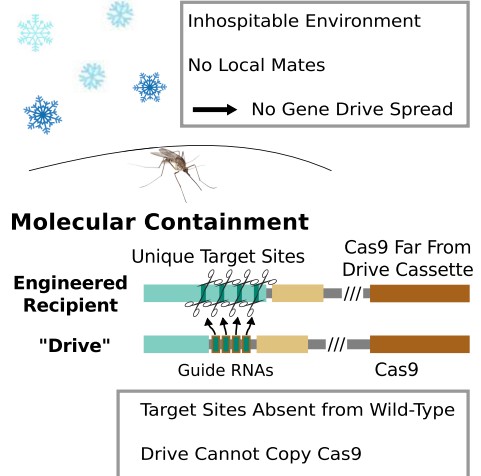

**Ecological Containment**

Inhospitable Environment

No Local Mates

⟶ No Gene Drive Spread

**Molecular Containment**

Engineered Recipient

Unique Target Sites

Cas9 Far From Drive Cassette

"Drive"

Guide RNAs          Cas9

Target Sites Absent from Wild-Type

Drive Cannot Copy Cas9

⟶ No Gene Drive Spread

**Figure 8**. Containment strategies and ecological risk. *Ecological containment* involves building and testing gene drives in geographic areas that do not harbor native populations of the target species. For example, most gene drive studies involving tropical malarial mosquitoes have been conducted in temperate regions in which the mosquitoes cannot survive or find mates. *Molecular containment* ensures that the basic requirements for drive are not met when mated with wild-type organisms. True drives must cut the homologous wild-type sequence and copy both the gene encoding Cas9 and the guide RNAs. Experiments that cut transgenic sequences absent from wild populations and copy either the gene encoding Cas9 or the guide RNAs - but not both - should be quite safe. Ecological or molecular containment should allow basic research into gene drive effectiveness and optimization to be pursued with negligible risk. *Figure 8—figure supplement 1* categorizes these and many other possible experiments according to estimated risk.

The following figure supplement is available for figure 8:

**Figure supplement 1**. Estimated ecological risk of experiments during RNA-guided gene drive development.

societal review and consent. As self-propagating alterations of wild populations, RNA-guided gene drives will be capable of influencing entire ecosystems for good or for ill. As such, it is imperative that all research in this nascent field operate under conditions of full transparency, including independent scientific assessments of probable impacts and thoughtful, informed, and fully inclusive public discussions.

The decision of whether or not to utilize a gene drive for a given purpose should be based entirely on the probable benefits and risks of

that specific drive. That is, each drive should be judged solely by its potential outcomes, such as its ability to accomplish the intended aims, its probable effects on other species, the risk of spreading into closely related species by rare mating events, and impacts on ecosystems and human societies. As scientists developing these technologies, it will be our responsibility to make all empirical data and predictive models freely available to the public in a transparent and understandable format. Above all else, we must openly share our level of confidence in these assessments as we determine how best to proceed.

## Discussion

The potentially widespread implications of RNA-guided gene drives demand a thoughtful and collected response. Numerous practical difficulties must be overcome before gene drives will be in a position to address any of the suggested applications. Many of our proposals and predictions are likely to fall short simply because biological systems are complex and difficult to engineer. Even so, the current rate of scientific advancement related to Cas9 and the many outcomes accessible using the simplest of gene drives suggest that molecular biologists will soon be able to edit the genomes of wild populations, reverse or update those changes in response to field observations, and perhaps even engage in targeted population suppression.

What criteria might we use to evaluate an RNA-guided gene drive intended to address a given problem? There are compelling arguments in favor of eliminating insect-borne human diseases, developing and supporting more sustainable agricultural models, and controlling environmentally damaging invasive species. At the same time, there are valid concerns regarding our ability to accurately predict the ecological and human consequences of these interventions. By bringing these possibilities before the scientific community and the public prior to their realization in the laboratory, we hope to initiate transparent, inclusive, and well-informed discussions concerning the responsible evaluation and application of these nascent technologies (*Oye et al., 2014*).

## Acknowledgements
We thank the participants of the MIT/Woodrow Wilson Center workshop 'Creating a Research Agenda for the Ecological Implications of Synthetic Biology' for sharing their thoughts concerning the potential impacts of this technology on various ecosystems (*Drinkwater et al., 2014*). We are grateful to S Lightfoot, K Oye, JP Collins, and M Wilkinson for critical reading of the manuscript, and to F Gould, T Wu, J Braff, and members of the Church and Oye laboratories for helpful discussions. KME was supported by a Wyss Technology Development Fellowship.

## Funding

| Funder | Grant reference number | Author |
|---|---|---|
| Wyss Institute for Biologically Inspired Engineering | Technology Development Fellowship | Kevin M Esvelt |
| Wyss Institute for Biologically Inspired Engineering | Synthetic Biology Platform funds | Kevin M Esvelt, Andrea L Smidler |

The funders had no role in study design, data collection and interpretation, or the decision to submit the work for publication.

## Author contributions
KME, Conceived of the study, performed the analysis, designed new drive types, drafted and revised the article; ALS, Designed new drive types, drafted and revised the article; FC, GMC, Revised the article

**Kevin M Esvelt** Synthetic Biology Platform, Wyss Institute for Biologically Inspired Engineering, Harvard Medical School, Boston, United States
http://orcid.org/0000-0001-8797-3945
**Andrea L Smidler** Synthetic Biology Platform, Wyss Institute for Biologically Inspired Engineering, Harvard Medical School, Boston, United States; Department of Immunology and Infectious Diseases, Harvard School of Public Health, Boston, United States
http://orcid.org/0000-0002-3281-1161
**Flaminia Catteruccia** Department of Immunology and Infectious Diseases, Harvard School of Public Health, Boston, United States; Dipartimento di Medicina Sperimentale e Scienze Biochimiche, Università degli Studi di Perugia, Terni, Italy
**George M Church** Synthetic Biology Platform, Wyss Institute for Biologically Inspired Engineering, Harvard Medical School, Boston, United States
*Competing interests:* KME: Filed for a patent concerning RNA-guided gene drives. ALS: Filed for a patent concerning RNA-guided gene drives. GMC: Filed for a patent concerning RNA-guided gene drives. The other author declares that no competing interests exist.

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
