## [Decision Letter]

Thank you for sending your work entitled “Concerning RNA-Guided Gene Drives for the
Engineering of Wild Populations” for consideration at *eLife*. Your
article has been favorably evaluated by Detlef Weigel (Senior editor) and 3 reviewers,
one of whom is another Senior editor.

The Reviewing editor, the other reviewers, and the editorial leadership discussed their
comments before we reached this decision, and the Reviewing editor has assembled the
following comments to help you prepare a revised submission:

We appreciate that the overall purpose of the article is to alert the scientific
community to the new capabilities offered by the CRISPR nuclease Cas 9 RNA-guided gene
drives in engineering wild populations. You briefly review the concept of gene drives,
the CRISPR/Cas9 technology, and current research driving advancement of gene drive and
CRISPR technologies. Most of the manuscript focuses on “likely capabilities” of Cas9
mediated gene drives, including the theoretical uses for “sensitizing”, “unstable”,
“interacting”, and “immunizing” drives. You briefly state that these future applications
of Cas9 gene drives are associated with environmental, safety, and security concerns
that should be considered before any experimentation is conducted. You conclude that
Cas9 RNA-guided gene drives offer new capabilities and applications beyond what is
currently available, but the implications for the environment and society as a whole
should be considered. You do not, however, address any issues concerning a possible
deliberately harmful use of the technique; since it does not involve complex technology,
there is the possibility that it is applied outside the reaches of governmental or
societal regulation.

To encourage thoughtful review of the Cas9 technology as it continues to develop and be
applied in different ways, you should ensure that your manuscript is clear about which
technologies have been developed, which are currently under active investigation, and
which are yet to be developed. To better encourage public discussion on this complex
technology, you should describe the CRISPR/Cas9 technology more clearly, as the sections
describing the basic technology are difficult to read and visualize for non-specialists.
It would be helpful to define CRISPR, Cas9, the DNA repeat elements, their natural
functions (targeted cleavage of foreign DNA), and how they can be tailored to use as
gene editing drives in multi-cellular organizations. This basic description is important
because it is a new technology even for many molecular biologists, let alone for the
interested public, and without a clear understanding of the technology, the reader might
not be able to fully comprehend the societal implications (both benefits and risks) and
likelihood of future capabilities of the technology.

We also felt that there is the danger that without a strong, factual discussion of the
basic concepts along with what is currently possible and what has so far not been
demonstrated, the intended audience (public, broader science community, others with no
or different technical knowledge) might arrive at a conclusion that does not adequately
reflect current and near-future reality. When basic scientific topics are clearly
described, the responses tend to be much more thoughtful and measured. When the science
is not sufficiently described, all sorts of responses are elicited.

This article will be treated as a Feature article, so it will be clear on both the web
and PDF versions that it is not a Research article. This will be especially clear in the
PDF, which will have a two-column layout. It is planned to be published under the
heading “Emerging Technologies” and it is suggested to change the title into “Possible
applications and potential concerns related to the use of RNA-guided gene drives to
engineer wild populations”. The Abstract should be intelligible to a broad public. The
*eLife* editors offer to work with you to ensure that the title and
abstract accurately reflect the content of the article.

Supplementary notes cannot be used in the intended article format. Notes 2 and 4
describe important possible limitations of the technology that should be integrated into
the text to ensure that it is understood by the general reader that the technology is
not immediately ready for use and may even turn out to be not as powerful as it might
seem at present. Notes 1 and 3 could be moved into boxes. Please ensure in all passages
that a balanced discussion between possible benefits and (current) technological
limitations is maintained.

To avoid any possibly distracting discussion around this article, you are asked to be
particularly careful with citing previous relevant work when discussing specific issues
and ideas.

---

## [Author Response]

*We appreciate that the overall purpose of the article is to alert the scientific
community to the new capabilities offered by the CRISPR nuclease Cas 9 RNA-guided
gene drives in engineering wild populations. […] You conclude that Cas9 RNA-guided
gene drives offer new capabilities and applications beyond what is currently
available, but the implications for the environment and society as a whole should be
considered. You do not, however, address any issues concerning a possible
deliberately harmful use of the technique; since it does not involve complex
technology, there is the possibility that it is applied outside the reaches of
governmental or societal regulation*.

We hope to initiate discussion concerning how this potentially powerful technology can
be responsibly developed, evaluated, and used to solve major ecological problems in a
way that minimizes the risk of harmful consequences. This latter purpose is best served
by ensuring that control measures are developed concomitantly. We consequently devote
much of the manuscript to detailing reversal and immunization drives intended to undo
earlier genomic changes and protect populations against gene drives. To emphasize the
importance of these novel drive architectures in this revision, we have combined the
earlier figures describing these drives into a single larger and clearer Figure 5. We have also included boxes that
explicitly address whether gene drives could be used to alter populations of humans,
crops, or domesticated animals. The possibility that unauthorized gene drives might be
deliberately misused is more relevant to regulators and consequently is discussed in
greater detail in our related manuscript on preparedness and risk governance scheduled
for simultaneous publication in *Science*.

*To encourage thoughtful review of the Cas9 technology as it continues to develop
and be applied in different ways, you should ensure that your manuscript is clear
about which technologies have been developed, which are currently under active
investigation, and which are yet to be developed. To better encourage public
discussion on this complex technology, you should describe the CRISPR/Cas9 technology
more clearly, as the sections describing the basic technology are difficult to read
and visualize for non-specialists. It would be helpful to define CRISPR, Cas9, the
DNA repeat elements, their natural functions (targeted cleavage of foreign DNA), and
how they* can *be tailored to use as gene editing drives in
multi-cellular organizations. This basic description is important because it is a new
technology even for many molecular biologists, let alone for the interested public,
and without a clear understanding of the technology, the reader might not be able to
fully comprehend the societal implications (both benefits and risks) and likelihood
of future capabilities of the technology*.

*We also felt that there is the danger that without a strong, factual discussion
of the basic concepts along with what is currently possible and what has so far not
been demonstrated, the intended audience (public, broader science community, others
with no or different technical knowledge) might arrive at a conclusion that does not
adequately reflect current and near-future reality. When basic scientific topics are
clearly described, the responses tend to be much more thoughtful and measured. When
the science is not sufficiently described, all sorts of responses are
elicited*.

We agree that the inclusion of a section explaining the established use of Cas9 for
genome editing would greatly strengthen the paper. While the subject of CRISPR systems
and Cas9-mediated genome engineering is large enough for entire review articles, we have
done our best to provide a concise but thorough explanation in a new section at the end
of the Introduction. We briefly detail Type II CRISPR systems as requested, but devote
the bulk of the section to RNA-guided genome editing. We have additionally created a new
figure detailing the mechanism of genome editing that emphasizes the similarity of this
process to the homing observed in endonuclease gene drives.

*This article will be treated as a Feature article, so it will be clear on both
the web and pdf versions that it is not a Research article. This will be especially
clear in the pdf, which will have a two-column layout. It is planned to be published
under the heading “Emerging Technologies” and it is suggested to change the title
into “Possible applications and potential concerns related to the use of RNA-guided
gene drives to engineer wild populations”. The Abstract should be intelligible to a
broad public. The* eLife *editors offer to work with you to ensure
that the title and abstract accurately reflect the content of the
article*.

The Abstract and Introduction have been revised to more clearly explain the topic to a
general audience. We welcome the chance to emphasize in the article format that this
technology remains entirely theoretical, as our intent is to initiate discussion prior
to development of this technology in the laboratory. We are grateful to
*eLife* for establishing a new section on emerging technologies for
this purpose.

While clarifying public perception is in this case an overriding reason to classify this
as a Feature article rather than a Research article, we regret the implication that
theoretical analysis does not constitute scientific research. Many of the most respected
publications in the history of science consisted entirely of conceptual insights – some
of them without any mathematical or computational component.

*Supplementary notes cannot be used in the intended article format. Notes 2 and 4
describe important possible limitations of the technology that should be integrated
into the text to ensure that it is understood by the general reader that the
technology is not immediately ready for use and may even turn out to be not as
powerful as it might seem at present. Notes 1 and 3 could be moved into boxes. Please
ensure in all passages that a balanced discussion between possible benefits and
(current) technological limitations is maintained*.

Points relevant to the non-specialist reader that were previously covered in the Notes
have been incorporated into the text and new boxes if they were not already present.
Material of concern only to specialists has been transferred to new figure supplements:
Figure 4—figure supplement 2, Figure 4—figure supplement 3, and Figure 7—figure supplement 1. Figure 4—figure supplement 3 specifically details existing
technologies that could be applied to optimize RNA-guided gene drives and notes the
challenges involved in doing so.

We have endeavored to clarify that this technology remains theoretical and our
predictions are fallible throughout. Most notably, the last sentence of the Abstract
describes the technology as “currently theoretical” and the section “Building Functional
Gene Drives” has been prominently renamed “Will RNA-Guided Gene Drives Enable Us to Edit
the Genomes of Wild Species?” It goes on to explicitly state that we cannot know for
certain until we try. Combined with our cautionary statement in the Conclusion noting
that, “Numerous practical difficulties must be overcome before gene drives will be in a
position to address any of the suggested applications. Many of our proposals and
predictions are likely to fall short simply because biological systems are complex and
difficult to engineer...”, we think it unlikely that a reader will come away with a
mistaken impression.

*To avoid any possibly distracting discussion around this article, you are asked
to be particularly careful with citing previous relevant work when discussing
specific issues and ideas*.

We have endeavored to cite relevant works and additionally mention several major
contributors to the field of endonuclease gene drives by name.